# *Bifidobacterium lactis* and *Lactobacillus plantarum* Enhance Immune Function and Antioxidant Capacity in Cats through Modulation of the Gut Microbiota

**DOI:** 10.3390/antiox13070764

**Published:** 2024-06-25

**Authors:** Weiwei Wang, Hao Dong, Xiaohan Chang, Qianqian Chen, Longjiao Wang, Shuxing Chen, Lishui Chen, Ran Wang, Shaoyang Ge, Pengjie Wang, Yixuan Li, Siyuan Liu, Wei Xiong

**Affiliations:** 1Food Laboratory of Zhongyuan, Luohe 462300, China; bingzhi213608@163.com (W.W.); donghao@zyfoodlab.com (H.D.); damei844022248@gmail.com (X.C.); 2020920269@stu.haut.edu.cn (Q.C.); chengshuxing1@163.com (S.C.); chlishui@sina.com (L.C.); 2Key Laboratory of Precision Nutrition and Food Quality, Department of Nutrition and Health, China Agricultural University, Beijing 100083, China; longjiaowang2019@163.com (L.W.); wangran@cau.edu.cn (R.W.); geshaoyang@foxmail.com (S.G.); wpj1019@cau.edu.cn (P.W.); liyixuan@cau.edu.cn (Y.L.); siyuan.liu@cau.edu.cn (S.L.)

**Keywords:** probiotics, gut health, feline, immunomodulation, antioxidants

## Abstract

Gastrointestinal (GI) afflictions are prevalent among the feline population, wherein the intricacies of the gut microbiome exert a profound influence on their overall health. Alterations within this microbial consortium can precipitate a cascade of physiological changes, notably in immune function and antioxidant capacity. This research investigated the impact of *Bifidobacterium lactis* (*B. lactis*) and *Lactobacillus plantarum* (*L. plantarum*) on cats’ GI health, exploring the effects of probiotic supplementation on the intestinal ecosystem using 16S rRNA gene sequencing. The findings demonstrated a significant improvement in gut barrier function by reducing plasma concentrations of D-lactate (D-LA) by 30.38% and diamine oxidase (DAO) by 22.68%, while increasing the population of beneficial bacteria such as *Lactobacillus*. There was a notable 25% increase in immunoglobulin A (IgA) levels, evidenced by increases of 19.13% in catalase (CAT), 23.94% in superoxide dismutase (SOD), and 21.81% in glutathione peroxidase (GSH-Px). Further analysis revealed positive correlations between *Lactobacillus* abundance and IgA, CAT, and total antioxidant capacity (T-AOC) levels. These correlations indicate that *B. lactis* and *L. plantarum* enhance feline immune and antioxidant functions by increasing the abundance of beneficial *Lactobacillus* in the GI tract. These findings provide a foundation for probiotic interventions aimed at enhancing health and disease resistance in feline populations.

## 1. Introduction

The gastrointestinal (GI) tract plays a pivotal role in animal health, serving as the primary site for nutrient absorption and interactions with the immune system, and as a habitat for a complex microbial community [1,2]. Alterations in the composition of the gut microbiota can significantly affect metabolic pathways, influencing overall metabolic health and enhancing antioxidative processes [3]. Behary et al. [4] illustrated the critical role of specific gut microbiota species such as Bacteroides and Firmicutes in immune system regulation, particularly in non-alcoholic fatty liver disease and hepatocellular carcinoma, underscoring the complex interaction between gut microbes and host immunity. Another investigation by Hong et al. [5] explored the effects of aged gut microbiota species, including *Lactobacillus* and *Bifidobacterium*, on modulating host antioxidant systems. These findings demonstrate the many activities of the gut microbiota concerning animal health, emphasizing its critical contributions to immune response and antioxidant capacity.

The health of the GI tract of cats is vital in determining overall health because of its impact on nutrient absorption, immune function, and mental health. Recent research has emphasized the intricate relationship between a cat’s gut health and its overall physiological and psychological state. For instance, research has identified significant differences in the fecal microbiota between healthy cats and those suffering from inflammatory bowel disease or alimentary small cell lymphoma, suggesting a potential link between the composition of microbiota and the development of these diseases [6]. Additionally, a study exploring the impact of intestinal dysbiosis of varying severity on the gut’s microbial composition in felines has provided valuable insights into how microbiome diversity can influence disease resistance [7]. Moreover, the gut microbiota’s production of antioxidant enzymes plays a critical role in mitigating oxidative stress in cats, enhancing immune function and overall GI health [8].

Recent studies have elucidated the beneficial impacts of probiotics on the GI health of cats. When provided in sufficient quantities, these living bacteria are paramount in regulating the gut microbiota, bolstering barrier functions, and facilitating immunological responses. *Bifidobacterium lactis* (*B. lactis*) is known for its beneficial effects on the animal gut microbiome, enhancing GI health and systemic immunity. Research highlights the function of *B. lactis* in enhancing nutrient absorption, reducing GI disorders, and maintaining a balanced gut microbiota. Grubb et al. [9] demonstrated that *B. lactis* and bacteriophages decreased GI inflammation and altered gut microbial populations. Moreover, research on *B. lactis* BB-12 revealed its ability to alleviate disruption in the gut microbiota linked to obesity, indicating its potential to improve gut health and associated benefits [10]. *Lactobacillus plantarum* (*L. plantarum*) is recognized for stabilizing gut flora, enhancing digestive health, and boosting the immune system. Recent research has brought attention to the impacts of probiotics on animal health; for instance, Huang et al. [11] demonstrated that *L. plantarum* CCFM8610 regulates metabolic pathways and gene expression in germ-free mice, emphasizing the probiotic’s ability to modulate host health. Additionally, the study conducted by Yin et al. [12] documented that *L. plantarum* GX17 improves growth performance and intestinal health in poultry, reinforcing its potential as a beneficial probiotic. *L. plantarum* CCFM1143 can significantly alleviate chronic diarrhea symptoms by modulating inflammation and gut microbiota, thereby improving patients’ health status and quality of life [13].The findings mentioned above highlight the pivotal role of *B. lactis* and *L. plantarum* in maintaining microbial balance in the GI tract of animals, contributing to improved health outcomes.

Research on the effects of probiotics on cats is sparse, despite the considerable body of scientific literature dedicated to their health benefits in humans and laboratory animals. This research aimed to fill a knowledge gap regarding the effects of *B. lactis* and *L. plantarum* on feline intestinal health, considering the distinct dietary requirements and GI physiology of cats, which might affect the composition of the gut microbiota. The study focused on the regulation of gut microbiota by these probiotics, which have the potential to improve GI health in felines. This research also assessed alterations in the gut environment post probiotic administration, exploring the mechanisms underlying the beneficial effects of these probiotics by employing 16S rRNA gene sequencing. The knowledge acquired may result in customized dietary strategies that improve the feline’s GI health and overall quality of life.

## 2. Materials and Methods

### 2.1. Animals and Experimental Treatments

The international research animal use guidelines and the Regulations for the Administration of Affairs Concerning Experimental Animals in China were followed throughout. The experimental protocols were approved by the Institutional Animal Care and Use Committee of China Agricultural University (AW11104202-5-2). All experiments were conducted with the utmost care to minimize animal suffering.

Ten healthy cats, aged between three and four months, were evenly divided into two groups based on their weight and physical condition scores, with each group containing five cats housed individually. The groups were assigned as follows: a control group (Con, *n* = 5 per group) and a probiotics group (Pro, *n* = 5 per group), with the latter receiving an extra dietary supplement of *B. lactis* and *L. plantarum* at 10 mg/kg body weight, and both sourced from the Nutrition and Health Research Institute of China Agricultural University, with preservation numbers CGMCC No. 11560 and CGMCC No. 14000, respectively. Both groups received a baked meal that met Association of American Feed Control Officials (AAFCO, 2023) and National Research Council (NRC, 2006) nutritional standards.

The research protocol included the consistent administration of powdered dietary supplements. Before the trial, all cats were immunized and dewormed, and we abstained from using antibiotics and other treatments that altered the microbiota for a minimum of one month. The controlled environment was characterized by a light–dark cycle of 12 h, temperatures ranging from 18 to 26 °C, and relative humidity levels between 40% and 70%. The cat’s dwelling was maintained with twice-daily feces disposal and daily sanitization. Each feline was given 35 g of food twice daily and unrestricted access to water. After one week of acclimation, a 28-day research phase was commenced, during which data for body weight, physical condition, and hair and fecal quality were collected on days 0, 7, 14, 21, and 28 to evaluate the effects of the dietary treatments.

### 2.2. Sample Collection

A volume of 1.5 mL of blood was taken from the cephalic vein located in the forelimb of each cat on the 28th day. A volume of 0.5 mL was transferred from this source into a vacutainer containing heparin sodium in preparation for regular hematological analysis. The remaining 1 mL was placed into a 1.5 mL centrifuge tube. The plasma was separated from these samples by centrifuging them at 3000 rpm for 15 min at 4 °C; separated plasma was then stored at −20 °C for subsequent analyses. Simultaneously, the sterile paper was inserted into the litter boxes of all cats in order to collect fecal samples. A total of 6 g of fresh feces was meticulously collected from each cat using sterilized forceps and placed into 50 mL sterile centrifuge tubes, with extreme care taken to prevent any contact with extraneous surfaces or objects. These fecal specimens were preserved at −80 °C for subsequent microbiota study.

### 2.3. Physical Condition, Fecal, and Fur Scoring

The physical condition and fecal scoring were conducted weekly using Teng and Carciofi’s proposed methodologies [14,15]. An experienced investigator assessed the condition of the fur. The scores were categorized using a three-point scale: 3 indicates perfect, clean fur; 2 represents intermediate conditions; 1 corresponds to disheveled, scruffy fur.

### 2.4. Blood and Serum Parameters Testing

The serum parameters were measured using the enzyme-linked immunosorbent assay (ELISA) technique, including inflammatory indices (tumor necrosis factor-α (TNF-α), interleukin-2 (IL-2), interleukin-4 (IL-4), and interferon-γ (IFN-γ)) and antioxidant capacity indicators (plasma superoxide dismutase (SOD), glutathione peroxidase (GSH-Px), total antioxidant capacity (T-AOC) and catalase (CAT)). According to the manufacturer’s instructions, the levels of D-lactate (D-LA) and diamine oxidase (DAO) were also ascertained using ELISA. Separate quantifications of serum immunoglobulins (IgG, IgM, IgA) were performed using the turbidimetric technique. All assay kits were acquired from Shanghai Enzyme-linked Biotechnology Co., Ltd. (Shanghai, China). A complete blood count (CBC) was conducted using a fully automated hematology analyzer (Contec, Qinhuangdao, China). All the blood samples for these measurements were taken after 28 days of feeding.

### 2.5. 16S rRNA Gene Sequencing and Microbial Bioinformatics Analysis

Following the manufacturer’s instructions, the total microbial DNA from feces was extracted using the E.Z.N.A.^®^ feces DNA Kit (Omega Bio-tek, Norcross, GA, USA). DNA concentration and purity were measured using 1% agarose gels and NanoDrop™ 2000 spectrophotometers (Thermo Fisher Scientific, Waltham, MA, USA). The bacterial 16S rDNA (V3–V4) was amplified using primers 338F (5′-ACTCCTACGGGAGGCAGCAG-3′) and 806R (5′-GGACT ACHVGGGTWTCTAAT-3′) by an ABI GeneAmp^®^ 9700 PCR thermocycler (Applied Biosystems, Foster City, CA, USA). The PCR product was extracted from 2% agarose gel and purified using the AxyPrep DNA Gel Extraction Kit (Axygen Biosciences, Union City, CA, USA) following the manufacturer’s instructions, and quantified using Quantus™ Fluorometer (Promega Corporation, Madison, WI, USA). The paired-end sequencing of equimolar pools of purified amplicons was performed on an Illumina MiSeq PE300 platform (Illumina, San Diego, CA, USA) adhering to the standard protocols set by Majorbio Bio-Pharm Technology Co. Ltd. (Shanghai, China).

Demultiplexing, quality filtering with fastp version 0.20.0, and merging with FLASH version 1.2.7 were performed on the unprocessed 16S rRNA gene sequencing reads. UPARSE version 7.1 was utilized to cluster operational taxonomic units (OTUs) with a 97% similarity cutoff; chimeric sequences were identified and eliminated. Chao and Shannon estimates were made of the alpha diversity of the microbial community using QIIME 2 (Version 2020.6). Principal Coordinates Analysis (PCoA) illustrated the complexity and diversity of the bacterial communities’ structure using the Bray–Curtis with arithmetic average. Bacterial abundance was calculated at taxonomic levels ranging from phylum to species in the form of percent abundance. The difference in the dominant bacterial community among the groups was identified using a Kruskal–Wallis H test bar plot. Spearman’s correlation heatmaps were constructed using Spearman’s correlation coefficients between bacterial communities and serum immune factor. A Spearman correlation heatmap based on the Spearman correlation coefficients among the bacterial profiles and serum indices was produced using R software (version 2.15.3).

### 2.6. Statistical Analyses

The experimental findings were analyzed using IBM Statistical Package for Social Sciences (SPSS) software (version 22.0; SPSS Inc., Chicago, IL, USA). Student–Newman–Keuls multiple range tests were implemented to evaluate variation among treatments. Significance was established at *p* ≤ 0.05; the data are expressed as the mean ± standard error of mean (SEM). The graphics in this study were produced using GraphPad (GraphPad Prism, version 8.0.2, San Diego, CA, USA).

## 3. Results

### 3.1. Growth Performance

The effects of probiotics on the fecal scores, physical condition scores, body weights, and hair conditions of cats throughout the experiment are depicted in Figure 1. An insignificant difference was observed in the fecal scores between the two groups on day 0. On day 7, the cats in the Pro group had a lower fecal score than those in the Con group (*p* = 0.02). On days 14, 21, and 28, there were no significant variations in the fecal scores between the two groups of cats. However, the fecal scores of the Pro group were more concentrated around 2.5 points (Figure 1a). Throughout the experiment, there was no significant disparity in the mean physical condition scores between the two cat cohorts. After 28 days of probiotic feeding, each cat in the Pro group achieved the standard body condition score of 5 (Figure 1b). There were insignificant variations in the body weights of cats on days 0, 7, 14, 21, and 28 between the two treatments (Figure 1c). After 28 days of feeding, the Pro group’s hair condition score was significantly higher than that of the Con group (*p* = 0.03) (Figure 1d).

### 3.2. Complete Blood Count

Table 1 presents the effects of probiotics on cats’ CBC. There was an insignificant difference in CBC between the Con and Pro groups of cats.

### 3.3. Gut Barrier Function Parameters

Figure 2 illustrates the measured concentrations of D-LA and DAO, two crucial indicators of gut barrier integrity. D-LA levels were 0.79 mmol/L in the Con group but reduced to 0.55 mmol/L in the Pro group. Similarly, DAO concentrations dropped from 1.94 U/mL in the Con group to 1.50 U/mL in the Pro group. The analysis revealed significant reductions in the Pro group compared to the Con group, with decreases of 30.38% for D-LA and 22.68% for DAO (*p* ≤ 0.05).

### 3.4. Immunoglobulin Parameters

The immunological response was assessed by measuring the IgG, IgM, and IgA concentrations, as depicted in Figure 3. The IgG and IgM concentrations in the Pro group showed no significant differences compared to the Con group. Conversely, IgA concentrations were 1.36 g/L in the Con group and 1.70 g/L in the Pro group, representing a significant 25% increase in IgA levels in the Pro group compared to the Con group—a statistically significant difference (*p* = 0.004).

### 3.5. Antioxidant Parameters

The concentrations of CAT, SOD, GSH-PX, and T-AOC were quantified to evaluate the effects of probiotics on antioxidant markers in feline serum, as depicted in Figure 4. After a dietary intervention lasting 28 days, the Pro group exhibited significantly higher concentrations of CAT, SOD, and GSH-PX, measuring 48.35, 83.20, and 192.10 U/mL, respectively. In contrast, the Con group displayed 40.58, 67.12, and 157.72 U/mL, respectively. The Pro group exhibited significant increases in CAT, SOD, and GSH-PX concentrations in comparison to the Con group, which increased by 19.13% for CAT (*p* = 0.011), 23.94% for SOD (*p* = 0.0006), and 21.81% for GSH-PX (*p* = 0.01). There were no significant differences in the T-AOC concentrations between the Con and Pro groups.

### 3.6. Inflammatory Factors

As depicted in Figure 5, serum inflammatory factor concentrations were measured in Pro and Con groups, revealing the following: In the Pro group, IL-4 was measured at 7.94 pg/mL, IL-2 at 163.47 pg/mL, TNF-α at 42.94 pg/mL, and IFN-γ at 23.49 pg/mL. The Con group revealed IL-4 levels at 5.61 pg/mL, IL-2 at 189.96 pg/mL, TNF-α at 58.39 pg/mL, and IFN-γ at 29.15 pg/mL. The analysis indicated significant changes in the Pro group compared to the Con group, with an increase of 41.53% for IL-4 (*p* = 0.003) and decreases of 13.94% for IL-2 (*p* = 0.003), 26.46% for TNF-α (*p* = 0.01), and 19.45% for IFN-γ (*p* = 0.001).

### 3.7. Fecal Microbiota Composition

Figure 6 presents the results of an investigation into the composition and distribution of feline fecal microbiota utilizing 16S rRNA sequencing. A total of 513 OTUs were identified in the fecal samples from four separate groups of kittens. Of these, 131 OTUs were present in all groups, indicating the presence of core microbiota composition (Figure 6a). The examination of species richness among the group, as quantified by the Chao1 index, revealed statistically insignificant differences among the groups (*p* = 0.94) (Figure 6b). The Shannon index, which measures microbial community diversity, revealed insignificant variations in diversity across the groups (*p* = 0.75) (Figure 6c). However, PCoA highlighted distinct microbial community patterns for each group. Notably, a distinct differentiation was observed on the PCoA plot between the Pro28 and other groups (Figure 6d). The fecal microbiota of cats in the Pro28 group exhibited remarkable similarity after 28 days of probiotic administration, as evidenced by the more clustered arrangement of spots on the PCoA plot, suggesting a convergence of microbiota profiles compared to the initial state (day 0).

Figure 6e,f elucidate the relative abundance of bacterial taxa in feline fecal samples at the phylum and genus levels, respectively. At the phylum level, Firmicutes emerged as the predominant bacterial group across all examined groups, constituting an initial proportion of 74.60% in the Con0 group and 55.68% in the Pro0 group. Following 28 days, the relative abundance of Firmicutes in the Con28 group adjusted to 62.24%, whereas it increased significantly to 74.23% in the Pro28 group. Additionally, at baseline (Day 0), the Cro0 group was characterized by notable percentages of Actinobacteriota (9.81%), Bacteroidota (6.57%), and Campilobacterota (7.06%). In contrast, the Pro0 group displayed higher abundances of Actinobacteriota (20.99%) and Bacteroidota (21.30%), and a lower abundance of Campilobacterota (1.40%). Following 28 days of dietary intervention, the Pro28 group exhibited a significant increase in Firmicutes (74.23%) and a notable decline in Bacteroidota (2.57%) compared to Pro0. Furthermore, the proportion of Firmicutes in the Pro28 group surpassed that in the Con28 group (62.24%). At the genus level, the initial abundances in the Pro0 group were *Collinsella* (18.18%), *Peptoclostridium* (8.46%), *Megasphaera* (12.61%), *Lactobacillus* (2.94%), and *Prevotella* (13.95%). Comparing the Pro28 group to Con0, there was an increased abundance of *Lactobacillus* (27.28%) and *Peptoclostridium* (19.98%), and there was a drop in the abundance of *Megasphaera* (0.11%).

### 3.8. Differential Microbiota

High-dimensional biomarkers present in the microbiota of different treatment groups were identified via LEfSe analysis (Figure 7a). Significant alterations in microbial communities were detected using an LDA threshold of 2. In particular, the Pro28 group demonstrated a significant increase in the abundance of *Lactobacillus* and *Streptococcus* in comparison to the control groups at baseline (Con0) and after 28 days (Con28), as well as the initial probiotic group (Pro0). In contrast, *Marvinbryantia*, *Peptoclostridium*, *Slackia*, and *Peptostreptococcus* were much more abundant in the Con28 group.

A bar plot of the Kruskal–Wallis H test at the genus level demonstrated that Pro triggered several microbial alterations (Figure 7b). *Lactobacillus* and *Streptococcus* exhibited a significant increase in the Pro28 group, while *Peptoclostridium* showed a significant increase in the Con28 group. As demonstrated in Figure 7c–f, the proportion of sequences in *Lactobacillus* showed significant differences compared to the other three groups (*p* < 0.01). After 28 days of feeding, the abundance of *Peptoclostridium* in the Con group significantly increased (*p* < 0.05), while *Marvinbryantia* was significantly increased (*p* < 0.001) in Con28 compared to the other three groups. Furthermore, the abundance of *Anaerofustis* was significantly lower in the Con0 and Pro0 groups than in the Con28 group.

### 3.9. Correlation Analysis

This section elucidates the correlation patterns observed among the 20 most abundant bacterial taxa and various serum indices, as illustrated in Figure 8. *Lactobacillus* exhibits a strong positive correlation with IgA (*r* = 0.94), CAT (*r* = 1), and T-AOC (*r* = 0.94), but a significant negative correlation with IFN-γ (*r* = −0.94) and TNF-α (*r* = −0.94). *Megasphaera* has positive correlations with D-LA (*r* = 0.81) and TNF-α (*r* = 0.93), but negative correlations with IgA (*r* = −0.90), CAT (*r* = −0.84), SOD (*r* = −0.81), GSH-PX (*r* = −0.90), T-AOC (*r* = −0.90), and IL-4 (*r* = −0.81). *Fusobacterium* shows a positive correlation with D-LA (*r* = 0.93) and TNF-α (*r* = 0.81), but a negative correlation with SOD (*r* = −0.93) and IL-4 (*r* = −0.93). *Bifidobacterium* is positively correlated with CAT (*r* = 0.83), whereas *Peptococcus* is negatively correlated with IgG (*r* = −0.94).

## 4. Discussion

The simultaneous supplementation of these probiotics demonstrated varying effects on cats, with improvements in fecal scores observed at 7 days and improvements in hair condition observed at 28 days. However, no significant differences were observed in body weights and physical condition scores, aligning with other recent studies. Recent research supports our findings that probiotics improve cat hair condition and fecal quality. For instance, research has shown that the administration of probiotics and postbiotics has a harmless effect on oxidative stress and inflammatory biomarkers, underscoring the safety of these supplements [16]. Moreover, feeding probiotics increased the diversity of the intestinal microbiome, which suggests an improvement in gut health [17]. Our findings regarding the improvement in hair health scores are validated by Rodrigues et al. [18], who demonstrated that microencapsulating probiotics preserved their viability, potentially amplifying their beneficial impacts on skin and hair health. Studies have consistently shown a lack of substantial impacts on body weight, including observations that body weight scores remain unaltered when probiotics are administered to felines. This suggests that while probiotics can influence certain specific health parameters, their impact on weight might be limited [19].

Furthermore, Yang et al. [20] provided a comprehensive perspective on the health benefits of probiotics in pets, including cats. They highlighted the role of intestinal microbiota in general health and disease prevention. Finally, the enhanced hair condition observed in our Pro group could be associated with the findings reported by Li et al. [21], who observed that probiotic use was related to an enhancement in fecal antioxidants and a reduction in inflammatory markers, both of which could contribute to better skin and hair health. Recent research corroborates our experimental findings, suggesting that while probiotics may substantially alter body weight or physical condition scores in cats, they may improve specific attributes like hair quality and fecal health. These benefits are probably linked to the positive modulation of gut microbiota and their systemic impacts.

There were no statistically significant differences observed in the CBC parameters between the control group and the probiotic-treated group of cats. This finding aligns with previous research investigating probiotics’ impact on hematological parameters in felines. These results are supported by research by Fusi et al. [19], which indicates that supplementation with *L. acidophilus* exerted an insignificant effect on blood parameters, including red blood cells and white blood cells, compared to the control group. Similarly, it was found that probiotic administration has harmless effects on the systemic oxidative stress of felines, suggesting that probiotics may be considered safe with respect to hematological health [16]. The significant reduction in plasma D-LA and DAO concentrations in the probiotic group provides further evidence that probiotics improve intestinal barrier integrity via various mechanisms. This observation implies that the gut barrier function has been enhanced, which aligns with the outcomes of previous research, demonstrating that probiotics can fortify the gut epithelial barrier. In particular, probiotics have been revealed to upregulate the expression of tight junction proteins and increase mucin production, thus preventing the transfer of pathogens and inflammatory mediators into systemic circulation [22,23].

Furthermore, positive alterations in metabolomic profiles and fecal microbiota provide additional evidence of the beneficial effects of probiotics on intestinal health, supporting the notion that synbiotics might enhance gut barrier function [24]. The observed rise in IgA levels in the probiotic group may suggest enhanced mucosal immunity, as IgA is crucial for the immunological defense of mucosal surfaces. This is consistent with findings that highlight the overall beneficial effects of probiotics on the immune system and gastrointestinal health of pet animals, including the induction of immunoglobulin synthesis [20]. Additionally, it has been suggested that multistrain probiotics can promote the colonization of beneficial bacteria and improve immune status by elevating levels of microbiota-derived short-chain fatty acids and reducing inflammatory markers, potentially aligning with elevated IgA levels [21].

The substantial elevation in CAT, SOD, and GSH-PX levels observed in the probiotic group suggests a significant augmentation of the antioxidative defense mechanisms in felines; this corresponds with the results reported by Souza et al. [16], who observed enhanced oxidative stress in cats supplemented with probiotic. Furthermore, it has been documented that maternal supplementation with probiotics or synbiotics significantly increases CAT and GSH-Px activities in offspring, implying that probiotics might also enhance the antioxidant capacity of the offspring during development [25]. Moreover, research highlights the direct antioxidative properties of specific probiotic strains, supporting the notion that probiotics can play a significant role in controlling oxidative stress by modulating enzymatic antioxidants in animals [26].

This is further reinforced by studies demonstrating that specific probiotic strains alleviate oxidative stress in hepatocytes and various animal models, thus mitigating oxidative damage [27]. Additionally, the role of probiotics in enhancing the antioxidant defenses through upregulating key signaling pathways involved in oxidative stress response in animal models is well documented, affirming the beneficial impacts of probiotics on animal health [28]. These findings underscore the broad applications of probiotics in health and disease management, highlighting their crucial role in preventing oxidative-stress-related diseases by maintaining efficient antioxidant mechanisms.

The observed reduction in pro-inflammatory cytokines such as TNF-α and IFN-γ, along with a rise in IL-4 in the group receiving probiotic supplements, provides evidence of anti-inflammatory benefits in felines. This is supported by findings showing that multistrain probiotics can improve the immunological health of cats by modulating inflammatory and antioxidant markers [21]. Similar outcomes were observed in studies indicating that human probiotic supplements significantly improved antioxidant and inflammatory markers, suggesting that analogous benefits might occur in animals [29]. Furthermore, other research outlines probiotics’ broad antioxidative and anti-inflammatory potential, which could explain the lower levels of inflammatory cytokines in probiotic-supplemented cats, potentially achieved by modifying the gut microbiota and systemic immune responses [30].

Probiotics influence the composition and function of the gut microbiota, fostering a beneficial microbial habitat that promotes gut health and systemic immunity. The restoration of balance in the gut microbiota by probiotics can enhance short-chain fatty acid production, suppressing the pathogenic bacteria and promoting the growth of commensal bacteria, thereby improving gut barrier function and immunoregulation [31].

The distinct clustering identified in the Pro28 group within the PCoA suggests that probiotic supplementation causes significant shifts in the fecal microbiota composition, promoting a convergence toward a healthier or more stable microbiome profile in felines. This observation is consistent with a recent study on shelter kittens, which found that probiotic supplementation significantly altered fecal microbiota composition. In this randomized, placebo-controlled trial, the probiotic *Enterococcus hirae* reduced diarrhea incidence and led to distinct microbiota changes, as shown by PCoA. The probiotic group exhibited a healthier microbiota profile with increased bacterial diversity and stability, indicating beneficial modulation of the gut environment [32]. Further supporting this observation, recent findings demonstrate that a multistrain probiotic positively alters the gut microbiota in cats, resulting in increased levels of beneficial short-chain fatty acids and decreased inflammatory markers [21].

Additionally, the Pro28 group exhibited an increase in Firmicutes and a decrease in Bacteroidota. Similar taxonomic shifts resulting from probiotic supplementation were noted, suggesting improvements in microbial metabolism and gut health [17]. The present study demonstrates the enrichment of beneficial genera such as *Lactobacillus* and *Streptococcus* following probiotic supplementation, consistent with findings by Jang et al. [33], highlighting the capacity of feline-originated probiotics to modulate gut microbiota, improve immune responses, and optimize clinical health parameters in cats.

Additional evidence suggests that probiotics may influence gut flora, as shown by studies indicating that administering synbiotics to cats could substantially impact the composition and metabolomic profiles of fecal microbiota. This suggests that probiotics could potentially assist in restoring and sustaining a healthy gut microbiota in felines that have undergone antibiotic treatment [24]. Furthermore, the impact of dietary interventions, such as probiotics, on fecal microbiota, metabolites, and immune markers in felines is examined, providing additional support for the notion that probiotics are indispensable for the gut health and general well-being of felines [34]. These findings suggest that probiotics substantially impact the fecal microbiota composition, which may result in healthier gut microbiota profiles, and thus, better health outcomes for cats.

The correlation patterns discovered among several serum indices and the twenty most common bacterial taxa provide valuable insights into the possible interactions between the gut microbiota and the host’s immune responses. A high correlation between *Lactobacillus* and IgA, CAT, and T-AOC indicates that it may have a protective function in antioxidative responses and immunological regulation. The study of Suchodolski [35], which highlighted the impact of the gut microbiota on immune system dynamics in animals under healthy and sick conditions, provides more support for this result. Multiple associations between *Megasphaera* and biomarkers such as D-LA and TNF-α suggest it is involved in a complicated interplay with inflammatory mechanisms. These results correspond to Schluter et al. [36], who established a noteworthy correlation between gut microbiota composition and systemic immune cell dynamics in humans, proposing that analogous mechanisms may operate in cats. *Fusobacterium*’s potential involvement in promoting oxidative stress and inflammatory responses is underscored by the negative correlation it exhibits with SOD and IL-4. This notion was additionally investigated by Meazzi et al. [37] in their research into the interactions between the gut microbiota and feline coronavirus. The positive link between *Bifidobacterium* and CAT indicates that it may be of benefit in controlling oxidative stress, which corroborates the results of Lyu et al. [38], who discovered that *Bifidobacterium* promotes the health of cats by modulating the gut microbiota. Finally, the correlation between *Peptococcus* and IgG levels suggests the possibility of immunomodulatory properties, which might be a component of more significant microbial impacts on the immune system of the host, as elaborated by Zhang et al. [39] in their investigation encompassing gut microbiota, metabolome, and immune factors. The studies above contribute to advancing knowledge regarding the intricate relationship between gut microbiota and host health, thereby underscoring the potential of therapeutic interventions targeting gut microbiota in feline health.

## 5. Conclusions

The results of this study provide clear evidence that the simultaneous administration of *B. lactis* and *L. plantarum* to felines substantially enhances their intestinal health. The probiotics have been demonstrated to positively modulate the composition of gut microbiota, leading to improved gut barrier functions, as evidenced by reduced plasma D-LA and DAO levels. This modulation markedly increased beneficial bacteria populations such as *Lactobacillus*, facilitating a robust immunomodulatory response, as evidenced by elevated immunoglobulin A levels and enhanced antioxidant parameters like catalase and superoxide dismutase. In addition, the study highlights the importance of these probiotics in supporting the GI and systemic immune systems of feline hosts, thus enhancing their general well-being. Consequently, probiotics should be incorporated into dietary approaches that aim to improve feline health, as the results indicate that they have substantial therapeutic potential in managing GI health in cats. The insights derived from this research could lead to innovative approaches in the dietary management of cats, improving their immune functions and stabilizing gut microbiota to enhance their quality of life.

## Figures and Tables

**Figure 1 antioxidants-13-00764-f001:**
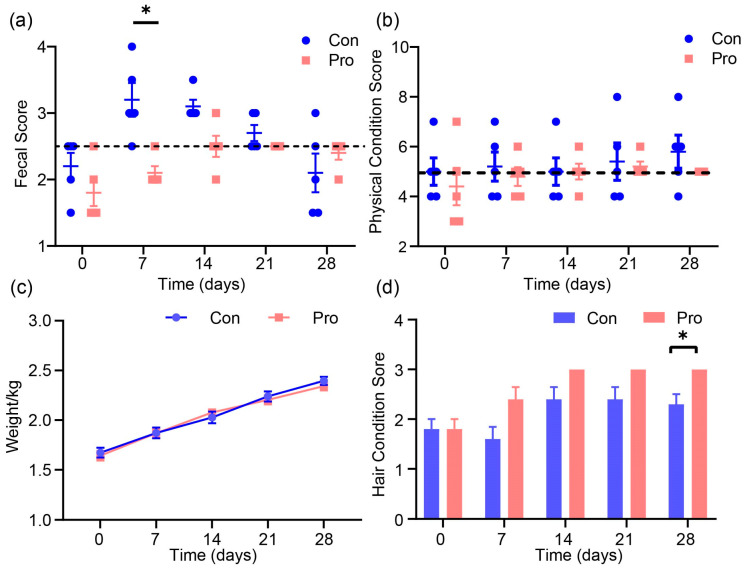
Effects of probiotics on growth performance in cats: (**a**) fecal score, (**b**) physical condition score, (**c**) body weight, and (**d**) hair condition score. A significant difference between the Probiotic (Pro) and Control (Con) groups at the same feeding time is denoted by an asterisk (* *p* ≤ 0.05) according to the paired-sample *t*-test. Values are expressed as means ± SEM, *n* = 5 per group.

**Figure 2 antioxidants-13-00764-f002:**
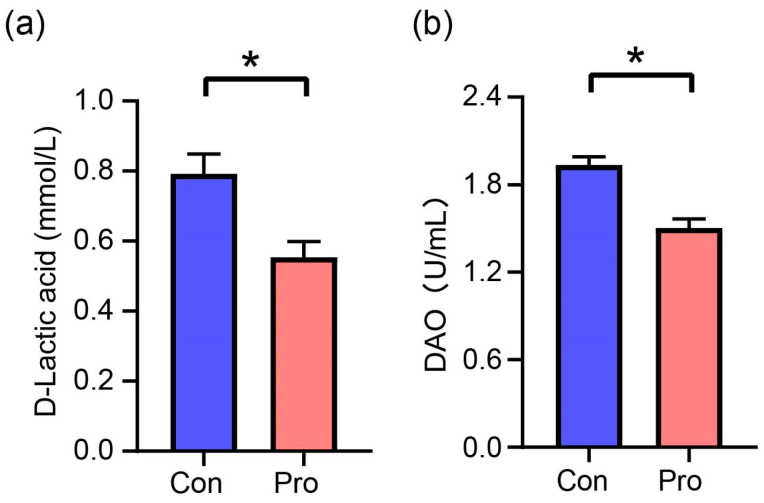
Effects of probiotics on plasma intestinal barrier function parameters in cats. (**a**) D-LA, D-lactate, and (**b**) DAO, diamine oxidase. A significant difference between Pro and Con groups at the same feeding time is expressed as * *p* ≤ 0.05 according to the paired-sample *t*-test. The values are expressed as means ± SEM, *n* = 5 per group.

**Figure 3 antioxidants-13-00764-f003:**
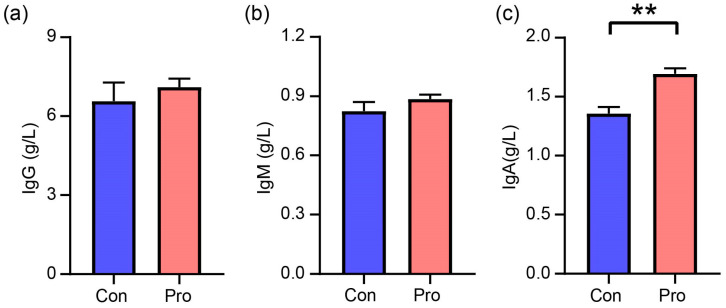
Effects of probiotics on immunoglobulin parameters in cats. (**a**) IgG, immunoglobulin G, (**b**) IgM, immunoglobulin M, and (**c**) IgA, immunoglobulin A. A significant difference between the Pro and Con groups throughout the same feeding time is expressed as ** *p* ≤ 0.01 according to the paired-sample *t*-test. The values are expressed as means ± SEM, *n* = 5 per group.

**Figure 4 antioxidants-13-00764-f004:**
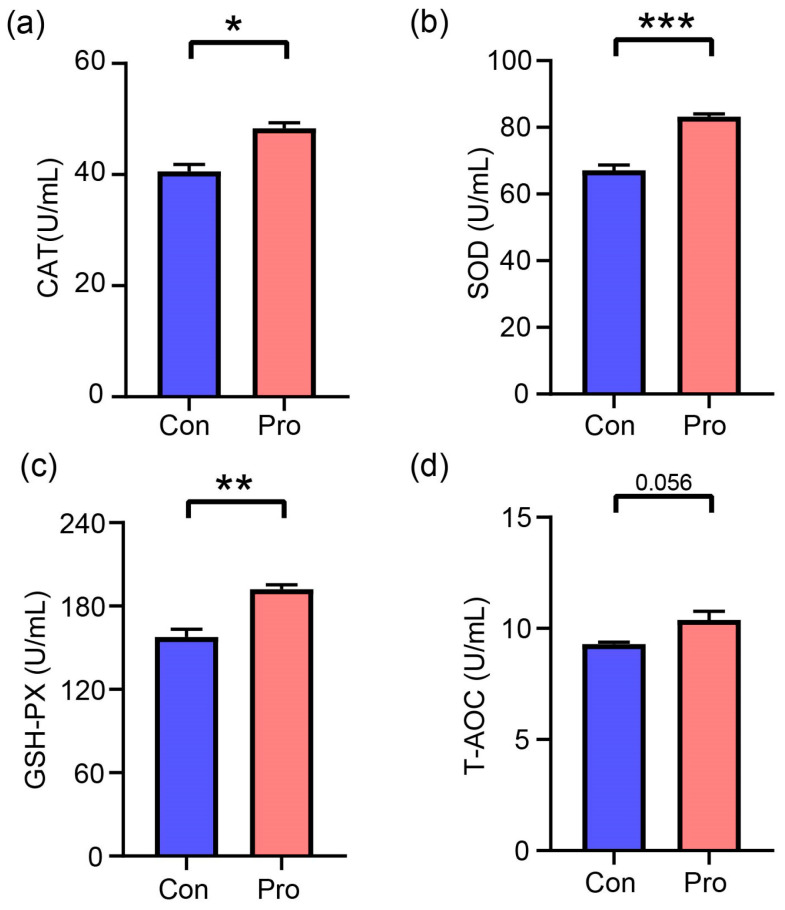
Effects of probiotics on antioxidant parameters in cats. (**a**) CAT, catalase, (**b**) SOD, superoxide dismutase, (**c**) GSH-PX, glutathione peroxidase, and (**d**) T-AOC, total antioxidant capacity. A significant difference between Pro and Con groups throughout the same feeding time is expressed as * *p* ≤ 0.05, ** *p* ≤ 0.01, and *** *p* ≤ 0.001 according to the paired-sample *t*-test. The values are expressed as means ± SEM, *n* = 5 per group.

**Figure 5 antioxidants-13-00764-f005:**
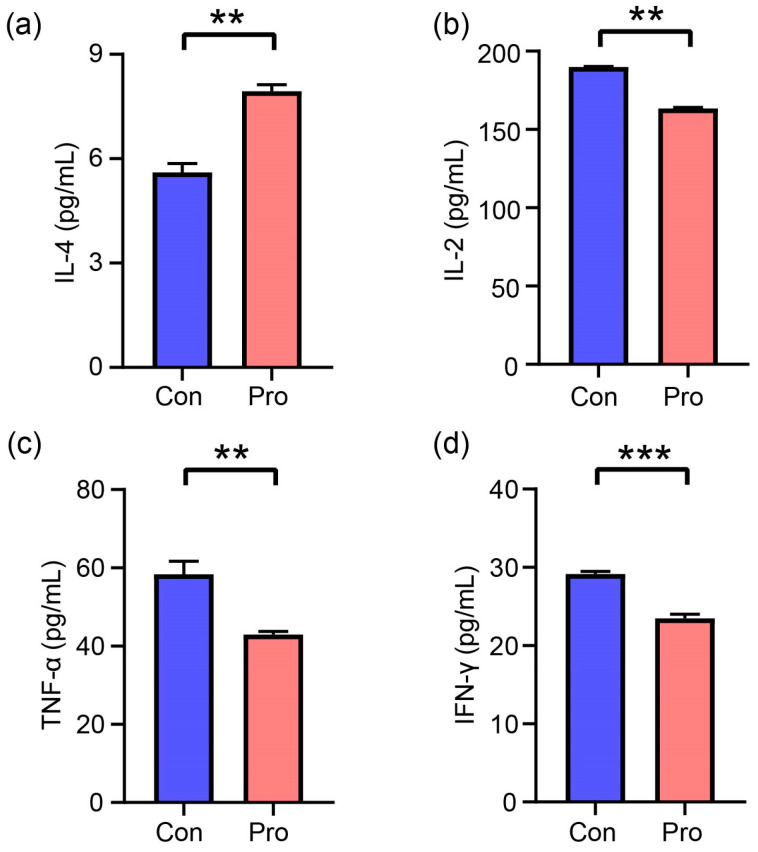
The effects of probiotics on plasma inflammatory parameters in cats. (**a**) IL-4, interleukin-4, (**b**) IL-2, interleukin-2, (**c**) TNF-α, tumor necrosis factor-α, and (**d**) IFN-γ, interferon-γ. A significant difference between Pro and Con groups at the same feeding time is expressed as ** *p* ≤ 0.01 and *** *p* ≤ 0.001 according to the paired-sample *t*-test. The values are expressed as means ± SEM, *n* = 5 per group.

**Figure 6 antioxidants-13-00764-f006:**
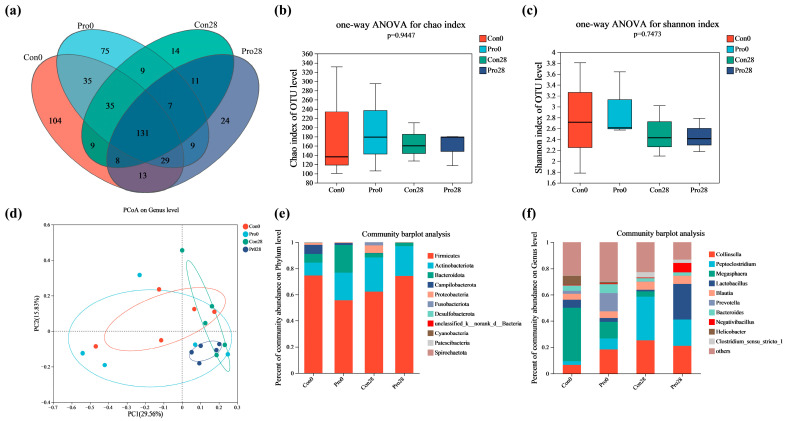
Effects of probiotics on the composition of fecal microbiota in cats. (**a**) Venn; (**b**) Chao index; (**c**) Shannon index; (**d**) Principal Coordinates Analysis; (**e**,**f**) Phylum and genus level of bacterial, respectively. The values are expressed as means ± SEM, *n* = 5 per group.

**Figure 7 antioxidants-13-00764-f007:**
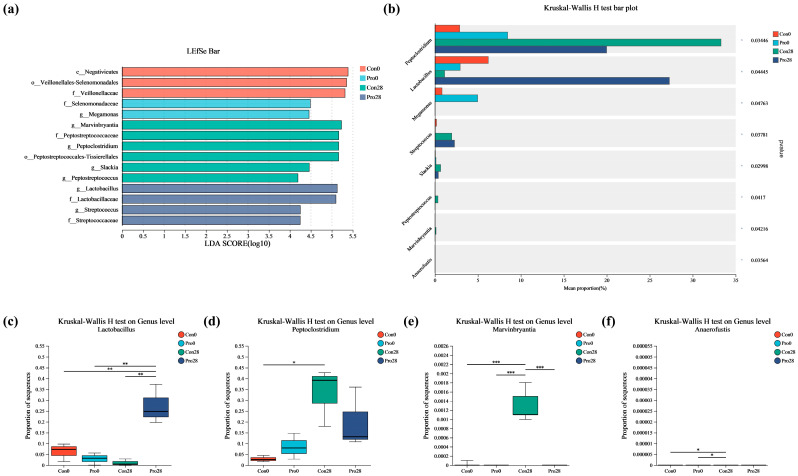
Effect of probiotic supplementation on feline gut microbiota composition. (**a**) LEfSe (linear discriminant analysis effect size) bar plot; (**b**) Kruskal–Wallis H test bar plot; panels (**c**–**f**) depict the relative abundance of key microbial genera: (**c**) *Lactobacillus*, (**d**) *Peptoclostridium*, (**e**) *Marvinbryantia*, and (**f**) *Anaerofustis*. Statistical significance: * *p* ≤ 0.05, ** *p* ≤ 0.01, and *** *p* ≤ 0.001. Data are represented as mean ± SEM based on a sample size of *n* = 5 per group.

**Figure 8 antioxidants-13-00764-f008:**
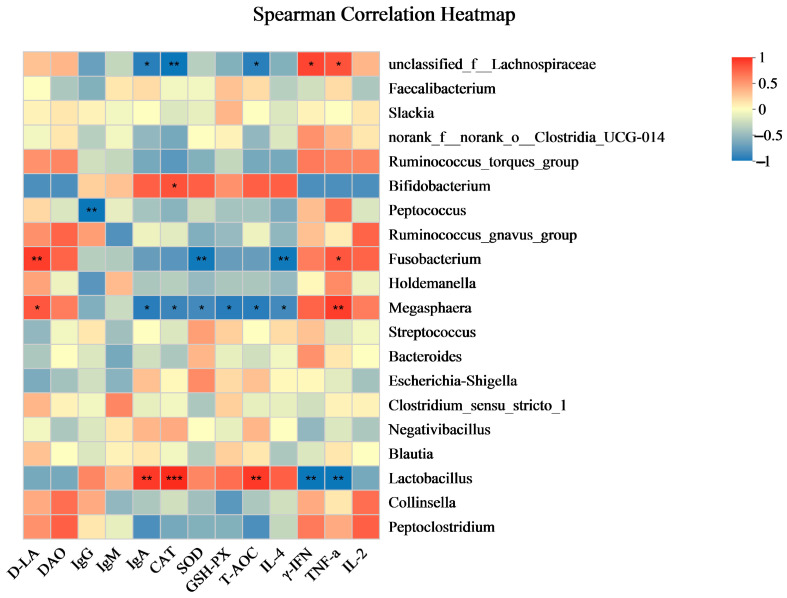
Spearman’s correlation heatmap depicting the associations between bacterial profiles and serum indices. Correlations are color-coded, with red representing positive associations and blue representing negative ones. The significance levels are * *p* ≤ 0.05, ** *p* ≤ 0.01, and *** *p* ≤ 0.001. Data are presented as mean ± SEM, with a sample size of *n* = 5 per group.

**Table 1 antioxidants-13-00764-t001:** The influence of probiotics on cat complete blood count (means ± SEM, *n* = 5 per group).

Item	Con	Pro	*p*-Value
Total White Blood Cells (10^9^/L)	24.76 ± 3.87	19.42 ± 4.00	0.294
Lymphocyte Ratio (%)	50.58 ± 6.11	48.96 ± 6.33	0.654
Intermediate Cell Ratio (%)	10.92 ± 0.52	11.34 ± 0.84	0.721
Granulocyte Ratio (%)	38.46 ± 5.99	39.30 ± 5.62	0.791
Lymphocytes (10^9^/L)	12.80 ± 2.77	11.30 ± 2.75	0.584
Intermediate Cells (10^9^/L)	2.72 ± 0.42	2.36 ± 0.40	0.511
Granulocytes (10^9^/L)	9.34 ± 1.53	7.86 ± 1.22	0.444
Total Red Blood Cells (10^12^/L)	7.92 ± 0.31	8.20 ± 0.76	0.790
Hemoglobin (g/L)	122.00 ± 6.94	127.40 ± 14.3	0.783
Hematocrit (%)	35.74 ± 1.66	36.54 ± 3.46	0.854
Mean Corpuscular Volume (fL)	45.28 ± 0.80	44.72 ± 1.32	0.705
Hemoglobin Content (pg)	15.32 ± 0.30	15.42 ± 0.39	0.831
Hemoglobin Concentration (g/L)	339.60 ± 5.11	346.60 ± 7.07	0.585
Red Cell Distribution Width SD (fL)	41.62 ± 1.91	35.34 ± 3.99	0.297
Red Cell Distribution Width CV (%)	19.38 ± 0.74	18.68 ± 0.87	0.385
Total Platelet Count (10^9^/L)	382.20 ± 31.20	364.40 ± 50.09	0.877
Mean Platelet Volume (fL)	11.58 ± 0.80	11.10 ± 1.20	0.763
Platelet Distribution Width (%)	10.60 ± 1.16	10.08 ± 1.30	0.813
Plateletcrit (%)	0.43 ± 0.06	0.45 ± 0.15	0.930
Platelet–Larger Cell ratio (%)	22.90 ± 2.03	21.32 ± 1.94	0.693

## Data Availability

The data generated from the study are clearly presented and discussed in the manuscript.

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
