# Peer review of "Bifidobacterium lactis and Lactobacillus plantarum Enhance Immune Function and Antioxidant Capacity in Cats through Modulation of the Gut Microbiota"

_antioxidants, 2024, doi:10.3390/antiox13070764_

Round 1

Reviewer 1 Report

General comments:

The present study aimed to evaluate the effect of simultaneous supplementation of two probiotics (Bifidobacterium lactis and Lactobacillus plantarum) on immune and oxidative status by gut microbiota modulation in cats. The Study is within the scope of the review and present novelty for this interesting subject. Overall, the manuscript was well structured and written regarding all sections and the conclusions are supported by results, despite the fact that Figures 3 to 5 don’t apport additional information to the text. However, some issues need be clarified or improved. The sample selection and group formation need be clarified, as well the source of the probiotics and statistical analyses for some repeated measures. Some words such as “notably”, “notable” and others should be avoided and replaced for more formal and objective terms. The limitation of the study (sample size and study design using a dual supplementation of probiotics without individual supplementation) should be reported and discussed. Despites these issues, we appreciate the full and equilibrate discussion of the findings of this study which were supported by appropriate literature.

Specific comments:

L69-79: “…improved GI inflammation…”? You mean positively modulated GI inflammation? I.e., decreased GI inflammation; I suggest to add “in humans.” at the end of the sentence.

L100-101: Please clarify if the cats were randomly selected for each group: what is the mean± weight of each one. According to Fig.  1(c) each group had the same body weight at the start of the experiment. Why they were categorized into two weight groups?

L129: Were conducted weekly

L133: And for time? Repeated measures: see Figure 1.

L183, 186: non-significant/ no significant. Please check the manuscript.

L103: You need to identify the source of the probiotics

L186-118: Predisposition? There is any tendency or differences between groups and time?

L190-192- I suggest to remove both sentences: P >0.05; thes difference in means is due to chance.

L192-194: “…dramatically when the feeding duration was extended, …”? I think that the next sentence is enough to report these results.

L199: There are not “Different capital letters (A–C) on the bars for…” in the graphs.

L203,219,232: Fig. 1(d) don’t present bar-; n=5 per group

L204: Notable? You mean “…significant difference (p = 0.004)…”.

L236-237: Please move this sentence to the end of the paragraph.

L324: Spearman correlations were not reported in M&M.

L342: The simultaneous supplementation of these probiotics.

L342: Noticeable?

L368: Insignificant?

L429: “in our group”. Please add a similar mention to your study when previously tou have a mention to other studies. Please check the manuscript.

L478: “… the simultaneous administration…”.

Author Response

Dear Reviewer,

Greetings!

Thank you for taking the time to read our manuscript and provide valuable feedback. Your insights are highly appreciated and have been carefully considered.

Please find our response to your comments attached. We are open to any further suggestions or discussions.

Wishing you success in your work and good health.

Reviewer 2 Report

Title: Bifidobacterium lactis and Lactobacillus plantarum enhance immune function and antioxidant capacity in cats through modulation of the gut microbiota

The manuscript requires reorganization, more details, grammar and writing check. 

The manuscript needs language editing.

More treatments should have been added, each probiotic should have been tested separately. 

If the difference is not significant, you can't claim its different from the other mean.

Use SEM instead of SD in all the tables and figures.

Title: Bifidobacterium lactis and Lactobacillus plantarum enhance immune function and antioxidant capacity in cats through modulation of the gut microbiota

The introduction requires reorganization. 

L20: GI

L20: who are the researchers? rewrite

L25: small letter for concurrently; rewrite, not clear

Reference 4 and 5: in what species.

L52: studies? you list only one study.

L61-62; 62-63; 64-65: reptation of the same ideas as previous paragraph. 

L105: year? is there NRC for cats? year?

L134: rewrite

L140: serum, small letter

L175 and elsewhere: put SEM instead of SD.

L184: was this significant? you made sound like its significant.

L190-191: rewrite, no need for, however.

L193: dramatically, replace

Fig 1 (D) what is the SD (SEM) at 14 and 28 d?

L222: not significant, not greater, rewrite.

L234-242: reorganize, discuss according to the figure order.

L352: their? 

The manuscript needs language editing.

Author Response

(The authors gave the same response as above.)

Round 2

Reviewer 2 Report

Thank you for providing revised version of the manuscript. You have answered all my points however I still see a slight problem with the design. It’s preferred to consult with a statistician for next experiment to have a better design. 

Most points we revised accordingly.